# Comparison of analgesic activities of aconitine in different mice pain models

**Jianhua Deng**[1,2☯], **Jiada Han**[1☯], **Jiahao Chen**[1], **Yanmin Zhang**[1], **Qiuju Huang**[1], **Ying Wang**[1], **Xiaoxiao Qi**[1], **Zhongqiu Liu**[1,3], **Elaine Lai-Han Leung**[3], **Dawei Wang**[2]*, **Qian Feng**[1]*, **Linlin Lu**[1,3]*

1 Joint Laboratory for Translational Cancer Research of Chinese Medicine of the Ministry of Education of the People's Republic of China, International Institute for Translational Chinese Medicine, Guangzhou University of Chinese Medicine, Guangzhou, Guangdong, China, 2 Shunde Hospital of Guangzhou University of Chinese Medicine, Guangzhou University of Chinese Medicine, Foshan, Guangdong, China, 3 State Key Laboratory of Quality Research in Chinese Medicine/Macau Institute for Applied, Research in Medicine and Health, Macau University of Science and Technology, Macau (SAR), China

☯ These authors contributed equally to this work.
* lllu@gzucm.edu.cn (LL); fengqian@gzucm.edu.cn (QF); david@gzucm.edu.cn (DW)

**Data Availability Statement:** All relevant data are within the paper.

**Funding:** This research was funded by the projects of National Natural Science Foundation of China [81874367], Natural Science Foundation for

## Abstract

Aconitine (AC) is the primary bioactive and secondary metabolite alkaloidin of Aconitum species which is accounted for more than 60% of the total diester-diterpenoid alkaloids in Aconite. To evaluate the analgesic effects of AC, 4 different pain models including hot plate assay, acetic acid writhing assay, formalin and CFA induced pain models were adopted in this study. In hot plate experiment, AC treatment at concentration of 0.3 mg/kg and 0.9 mg/kg improved the pain thresholds of mice similar to the positive drug aspirin at the concentration of 200 mg/kg (17.12% and 20.27% VS 19.21%). In acetic acid writhing experiment, AC significantly reduced the number of mice writhing events caused by acetic acid, and the inhibition rates were 68% and 76%. These results demonstrated that AC treatment revealed significant analgesic effects in both acute thermal stimulus pain model and chemically-induced visceral pain model. The biphasic nociceptive responses induced by formalin were significantly inhibited after AC treatment for 1h or 2h. The inhibition rates were 33.23% and 20.25% of AC treatment for 1h at 0.3 mg/kg and 0.9 mg/kg in phase I. In phase II, the inhibition rates of AC and aspirin were 36.08%, 32.48% and 48.82% respectively, which means AC showed similar analgesic effect to non-steroidal anti-inflammatory compounds. In the chronic CFA-induced nociception model, AC treatment also improved mice pain threshold to 131.33% at 0.3 mg/kg, which was similar to aspirin group (152.03%). Above all, our results verified that AC had obviously analgesic effects in different mice pain models.

## Introduction

Pain is a complex physiological and psychological activity of mammal [1]. Pain, both acute and chronic, remains a significant health problem, and it is also the most difficult challenges in medicine [2]. A recent epidemiological study showed that approximately 41.1% of adults in

Distinguished Young Scholars of Guangdong Province, China [2017A030306033], Guangdong Province Universities and Colleges Pearl River Scholar Funded Scheme (2016), Project of Educational Commission of Guangdong Province of China [2016KTSCX012], Pearl River Nova Program of Guangzhou, China [201710010108].

**Competing interests:** The authors have declared that no competing interests exist.

developing countries and 37.3% of adults in developed countries are suffering from pain due to illness or injury [3]. The Institute of Medicine reported that more than 20% of American adults experience chronic pain, which costs the United States more than 600 billion dollars annually, measured by health care usage and impact on quality of life [4]. Pain is not a unitary phenomenon, and its underlying biology can be collected by a series of receptors and channels [5]. Meanwhile, because pain seriously affects people's quality of life, there is an urgent requirement for better pain treatment. Due to fundamentally subjective and ethically self-limitation, it is a major challenge and unrealistic using human subjects to study pain. Therefore, preclinical animal models of pain are widely used.

Until now, an assortment of animal models has been developed to study the underlying mechanisms and therapies of pain. Undoubtedly, each model has its inherent advantages and limitations. As a result, different analgesics may exert different analgesic effects in the different models. Therefore, a basic understanding of mechanisms involved in pain models is necessary to choose the model most appropriate for examining analgesic properties of compounds. In our study, 4 commonly used pain models including hot plate assay, acetic acid writhing assay, formalin and complete freund's adjuvant (CFA) induced pain models were chosen to detect the analgesic activities of compounds.

Hot-plate model is the most commonly used model for measuring potential antinociceptive effects of compounds on acute thermal stimulus [6]. In this assay, animal was placed on a heated surface (55°C for mice or 52.5°C for rats) and the amount of time it took to lick rear paw was measured. Latency to respond to the heat stimulus represents the change of pain threshold. Acetic acid is typically used as noxious stimuli in visceral pain studies [7]. 0.6% v/v acetic acid was injected intraperitoneally into mice and the number of writhing events (stretching, retracting, or pressing the belly against the floor) was counted. The formalin assay is classified as a persistent pain model and produced biphasic nociceptive response [8]. Phase I is caused by pain fibers, especially C fibers and it starts immediately after formalin injection and lasts about 10 min. Phase II typically begins about 15 min after formalin injection and continues for ≥60 min which is mediated by peripheral inflammation. As a result, non-steroidal anti-inflammatory agents (NSAIDs) are ineffective in attenuating the phase I response, but are effective in attenuating phase II in formalin model. Subcutaneous injection of CFA is often used to model chronic inflammatory pain and the swelling it causes could persist for at least 7 days. This model is commonly used to detect the anti-inflammatory and analgesic effects of compounds [9, 10].

At present, traditional pain-management therapies mainly adopt the following 3 steps. Firstly, classic analgesics such as acetaminophen; Secondly, NSAIDs such as aspirin, ibuprofen [2, 11]; Finally, powerful analgesics like morphine [12]. Although these compounds are currently used in clinical, there are still some problems like not highly effective, significant side effects and so on [13, 14]. Admittedly, there is an urgent requirement for better pain therapies.

Aconitine (AC) is the primary bioactive and secondary metabolite alkaloidin of the Aconitum species [15]. AC accounted for >60% of the total diester-diterpenoid alkaloids in Aconite which is widely used in the treatment of rheumatoid Arthritis, cardiovascular disease and tumors [16]. Modern pharmacological studies have shown that Aconite has various biological activities such as analgesia, anti-arrhythmia, antitumor and so on [17–19]. However, it was still unknown whether AC had beneficial effects in treating pain. In the present study, hot plate assay, acetic acid writhing assay, formalin and CFA induced pain models were adopted to explore the in vivo analgesic effects of AC, and the results may provide experimental evidence for the clinical use of AC in treating pain.

## Material and methods

### Reagents

AC (purity > 98%) was purchased from Chengdu Mansite Pharmaceutical Co., Ltd. (Chengdu, China). Aspirin was purchased from Shanghai Yuanye Biotechnology Co., Ltd. (Shanghai, China). Glacial acetic acid was purchased from Tianjin Beilian Fine Chemicals Development Co., Ltd. (Tianjin, China). Formaldehyde was purchased from Thermo. Complete Freund's Adjuvant (CFA) was purchased from chondrex. AC was prepared into 3 mg/mL and 9 mg/mL stock solutions in DMSO and diluted to 0.03 mg/ml and 0.09 mg/ml with 0.9% saline. Aspirin was prepared into 20 mg/mL solutions in 0.15% sodium bicarbonate solution.

### Animals

Female C57BL/6 mice (18–20 g) were purchased from the Animal Center of Sun Yat-Sen University (License number: SCXK (Guangdong) 2016–0029; Guangzhou, China) and used for the experiments (total of 138 animals). Mice were housed at 23°C—25°C with relative humidity of 50% - 70% in a 12 h light/dark cycle, lights on at 7:00 am. Animals had free access to food and water. All animal experiments were approved by the Animal Ethics Committee of Guangzhou University of Chinese Medicine (IACUC, IITCM-01-20160703), in line with animal welfare, animal protection and ethical principles. The animals were grouped by simple randomization.

### Hot plate test

In the hot plate test, mice were randomly divided into four groups including high dosage AC (0.9 mg/kg), low dosage AC (0.3 mg/kg), aspirin (200 mg/kg) and control (0.9% saline), with 6 mice in each group. The drugs were administered orally. The temperature of hot plate was set at 55°C [6, 16, 20]. Response time for hindpaw withdraw of mice was recorded 30 min after the administration of test drugs. The base threshold of hindpaw withdraw latency was detected 30 min before the administration of test drugs. Pain threshold improvement rate = (latency after administration—base threshold) / base threshold × 100%.

### Acetic acid induced pain writhing experiment

In the pain writhing experiment, mice were randomly divided into four groups including high dosage AC (0.9 mg/kg), low dosage AC (0.3 mg/kg), aspirin (200 mg/kg) and control (0.9% saline), with 6 mice in each group. The drugs were administered orally. After 60 min of drug administration, all mice were intraperitoneally injected with 0.6% acetic acid in the dosage of 10 mL/kg. Then, writhing times of mice occurred within 15 min after the injection of acetic acid was recorded, and the inhibition rate was calculated [21]. Inhibition rate = (average writhing times of control group–average writhing times of drug treatment group) / average writhing times of control group × 100%.

### Formalin induced nociception assay

In the formalin induced nociception assay, mice were randomly divided into high dosage AC (0.9 mg/kg), low dosage AC (0.3 mg/kg), aspirin (200 mg/kg), model (0.9% saline) and control (0.9% saline) group, with 6 mice in each group. Single-dose drugs were administered orally. One or two hours after administration, 20 μL of 0.92% formaldehyde solution was subcutaneously injected into the dorsal surface of the right hindpaw of mice (control group was treated

with saline instead) [22, 23]. The cumulative time of paw licking during 0–5 min (first phase) and 15–60 min (second phase) after formalin treatment were recorded.

## CFA induced nociception assay

In CFA induced nociception assay, mice were randomly divided into high dosage AC (0.9 mg/kg), low dosage AC (0.3 mg/kg), aspirin (200 mg/kg), model (0.9% saline) and control (0.9% saline) group, with 6 mice in each group. 20 μL of CFA was subcutaneously injected into the dorsal surface of the right hindpaw of mice (control group was treated with 0.9% saline instead) [24]. After 48 hours, drug-treated groups were orally given AC or aspirin respectively once a day for seven consecutive days while the control group and model group were administrated with 0.9% saline; The swelling of mice toes were observed on the 1st, 3rd, 5th, and 7th day, and the latency of paw withdrawal was measured by hot plate test [24].

## Statistical analysis

Data were expressed as the mean ± standard deviation (SD). Significant differences were analyzed using one-way analysis of variance (ANOVA) followed by Tukey's *post hoc* test for multiple comparisons by SPSS 20.0. Statistical significances were considered at $p < 0.05$.

# Results

## Aconitine increased the pain threshold of mice in hot plate model

Firstly, we used hot plate mice model to determine analgesic activity of AC. In hot plate test, the reaction time of mice in AC treatment groups was significantly elongated in comparison to control group. The pain thresholds of mice were increased by 17.12% and 20.27% after AC administration at 0.3 mg/kg and 0.9 mg/kg, respectively (Table 1). The maximum effect was observed at the highest dose (0.9 mg/kg) which showed a reaction time of 7.6 sec, whereas the standard drug Aspirin (200 mg/kg) showed a reaction time of 5.0 sec. Compared with the standard drug aspirin, 0.9 mg/kg of AC showed similar effect on the improvement of pain threshold (20.27% vs 19.21%). The results indicated that AC was effective to release the pain induced by thermal stimulation.

## Aconitine reduced the number of writhing in acetic acid-induced writhing model

We next examined the analgesic effect of AC in acetic acid-induced writhing test. We found that AC treatment produced significant ($p<0.01$) reduction in the number of writhing in

**Table 1. Analgesic activity of aconitine in hot plate test in mice.**

| Group | Reaction time / s | Pain threshold improvement rate |
|---|---|---|
| | 30 min | 30 min |
| Control | 3.0±1.35 | — |
| Aspirin (200 mg/kg) | 5.0±1.42 | 19.21%** |
| Low dosage AC (0.3 mg/kg) | 4.2±0.65 | 17.12%** |
| High dosage AC (0.9 mg/kg) | 7.6±1.42*** | 20.27%** |

Data represent mean ± SD ($n = 6$).

**$p < 0.01$, and

***$p < 0.001$ compared with the control group.

mice. At 0.3 and 0.9 mg/kg oral dose, the percent reduction of writhing was 68% and 76% respectively, as compared to the control group, whereas the standard drug aspirin (200 mg/kg) showed a reduction of 75% (Table 2). The above results revealed that aspirin and AC could significantly reduce the number of writhing in mice caused by acetic acid, and more importantly, 0.9 mg/kg AC treatment showed similar analgesic effects as 200 mg/kg aspirin treatment.

## Aconitine reduced paw-licking time in formalin induced nociception assay

To investigate whether AC is effective to inflammatory pain, we examined the response of animals to formalin after AC treatment. According to the previous study, the $T_{1/2}$ of AC in mice is about 2 hours [20]. Hence, we measured the antinociceptive activity of AC in formalin induced paw-licking mice model at 1 and 2 hours after AC treatment, respectively. Our results showed that after 60 minutes of treatment, AC could significantly reduce formalin-induced paw licking time of mice both in the first phase and second phase. In the first phase, compared to the model group, the inhibition rates were 33.23% and 20.25% of 0.3 mg/kg and 0.9 mg/kg AC, and 32.03% of 200 mg/kg aspirin (Fig 1A, $p<0.01$). In the second phase, the inhibition rates of 0.3 mg/kg AC, 0.9 mg/kg AC, 200 mg/kg aspirin were 36.08%, 32.48% and 48.82% respectively, which showed significant suppression effects on formalin-induced nociception. Similarly, when formalin was injected two hours after drugs treatment, the paw licking time was significantly shortened by AC and aspirin too. Most of all, 0.9 mg/kg AC decreased the duration of paw licking to 25.73 sec in the second phase, which was lower than 51.81 sec of aspirin group (Fig 1B, $p<0.05$). And the inhibition rate of 0.9 mg/kg AC (68.32%) was 2.6 folds higher than aspirin group (25.97%). These results suggest that AC can significantly inhibit the pain response induced by formaldehyde.

## Aconitine inhibited CFA-induced paw edema and thermal hyperalgesia of mice

Finally, CFA-induced thermal hyperalgesia was used to detect the inhibitory effect of AC on inflammatory pain. As shown in Fig 2A, saline was injected into the control group, the swelling of the plantar of mice was observed only on day 1; while in the model group, 7 days after injection of CFA, paw edema accompanied by pus and rupture of mice were still obvious. Under treatment with 0.3 mg/kg and 0.9 mg/kg AC, paw edema of mice was effectively suppressed (Fig 2A), indicating that AC relieved the inflammation caused by CFA. Meanwhile, hot plate experiment was used to detect the effect of AC on thermal hyperalgesia in mice. CFA induced marked thermal hyperalgesia, as evidenced by reduced paw withdrawal latency after injection for 1–7 days, compared with the control group (Fig 2B, p<0.05). This CFA-induced thermal hyperalgesia was inhibited by administration with AC at 0.3 mg/kg and 0.9 mg/kg once daily from the 1st to the 7th day 48 hours after CFA injection. These results indicate that AC could inhibit the pain response induced by CFA.

**Table 2. Analgesic activity of aconitine in acetic acid writhing test.**

| Groups | Average no. of writhing (within 15 min) | Inhibition rate |
|---|---|---|
| Control | 28.83±11.53 | / |
| Aspirin (200 mg/kg) | 7.17±3.66** | 75% |
| Low dosage AC (0.3 mg/kg) | 9.17±5.88** | 68% |
| High dosage AC (0.9 mg/kg) | 7±5.14** | 76% |

Data represent mean ± SD (n = 6).

**p < 0.01 compared with the control group.

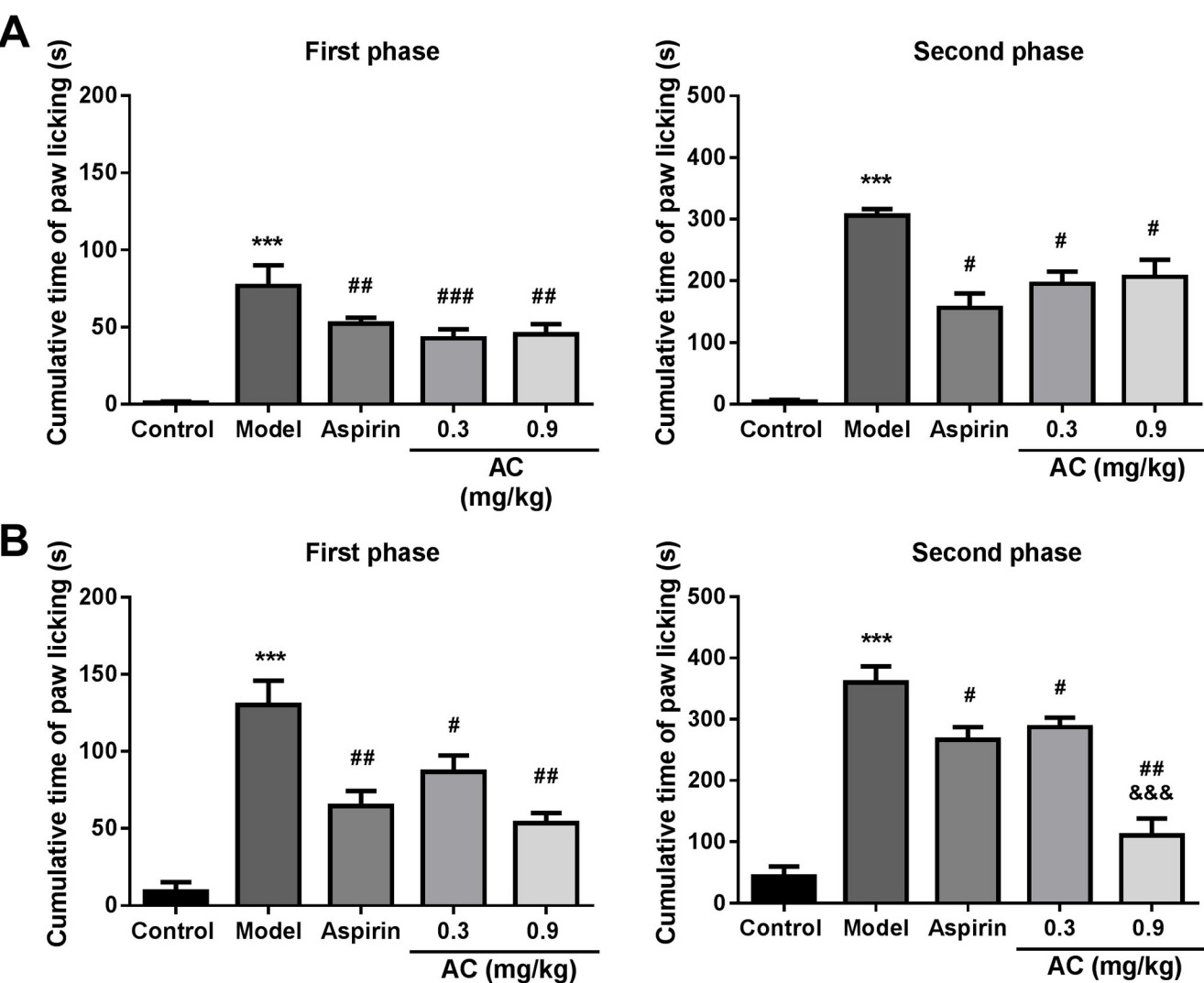

**Fig 1. Anti-nociceptive activity of aconitine in formalin induced nociception assay.** A, Different doses of AC or aspirin were administrated orally 1 hour before formalin injection. B, Different doses of AC or aspirin were administrated orally 2 hours before formalin injection. Data represent mean ± SD ($n = 6$). ***$p < 0.001$ compared with the control group; #$p < 0.05$, ##$p < 0.01$, ###$p < 0.001$ compared with the model group; &&&$p < 0.001$ compared with the aspirin group.

## Discussion

Aconitum has a long history of use in traditional Chinese medicine and it has a wide range of pharmacological effects. According to statistics, the frequency of aconite used in 500 known Chinese herbal formulas is 13.2% [25]. In modern clinic, aconite root is commonly used in the treatment of rheumatoid arthritis, rheumatic heart disease, acute myocardial infarction induced shock and a variety of other ailments [26, 27]. In 2020, the total consumption of Aconitum was more than 300 kg, about 25% of which is used for the treatments of pain, arthritis and other related diseases in Shunde Hospital of Guangzhou University of Chinese medicine. Recent evidences indicate that aconitine is the main chemical constituent and active ingredient of Aconitum [20, 28]. As far as we know, AC has a wide range of analgesic properties, but whether there are differences in the therapeutic effects of AC on different pain has not been reported. In the present study, we used four different pain mice models including hot plate

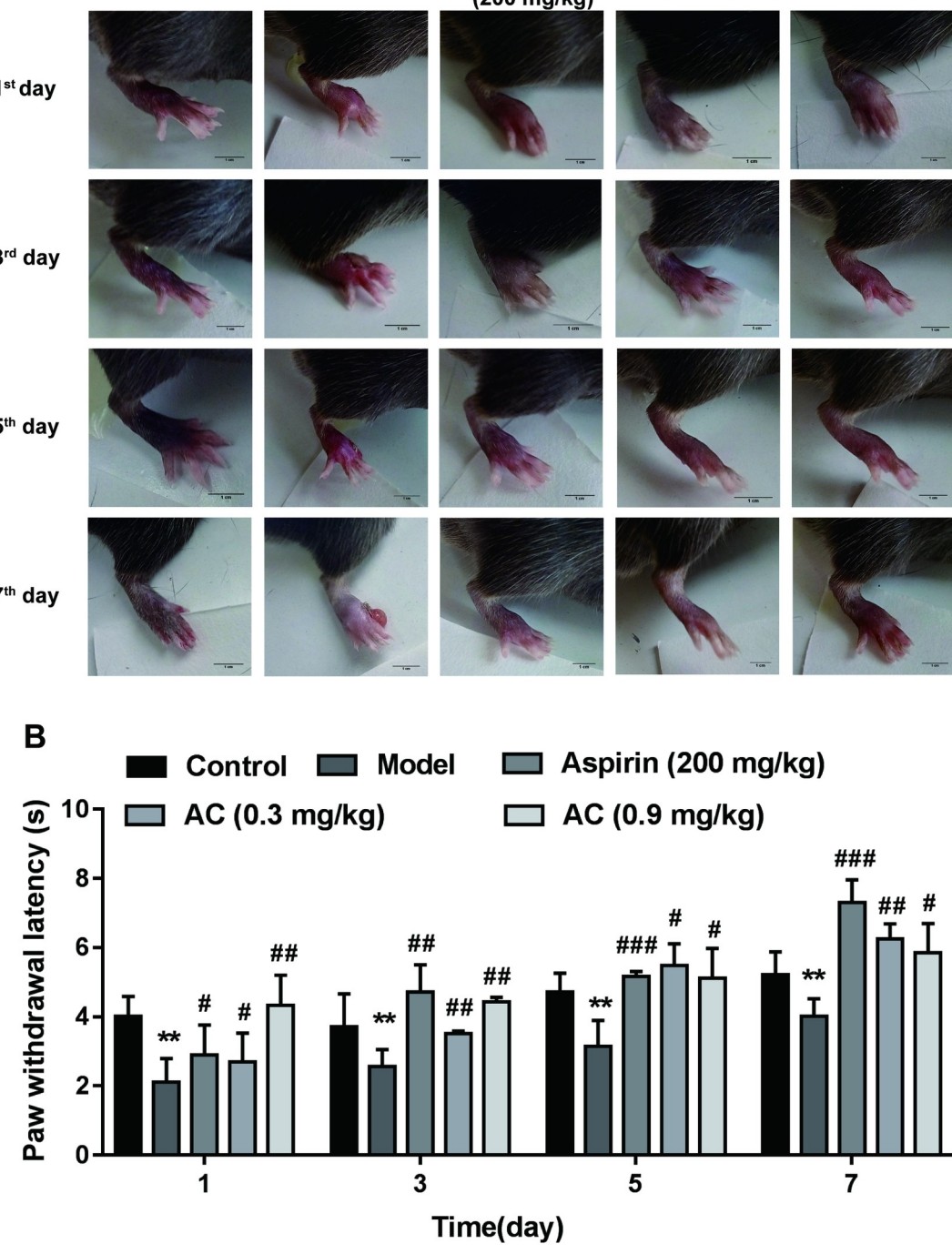

**Fig 2. Effects of aconitine on CFA-induced paw edema and thermal hyperalgesia in mice.** A, Paw swelling induced by CFA in mice. B, Paw withdrawal latency of mice was detected by hot plate test on the 1st, 3rd, 5th, and 7th day. Data represent mean ± SD ($n$ = 6). $^{**}p < 0.01$ compared with the control group; $^{#}p < 0.05$, $^{##}p < 0.01$, $^{###}p < 0.001$ compared with the model group.

test, acetic acid writhing test, formalin and CFA induced nociception assay to investigate the specific analgesic effects of AC. According to the previous study, mice rarely died after oral administration of 1.0 mg/kg AC for 21 days [29]. In order to avoid toxic symptoms, 0.3 mg/kg

Table 3. Analgesic activities of aconitine on different pain models of mice.

| Group | Acetic acid writhing test (inhibition rate, %) | Hot plate test (Pain threshold improvement rate, %) | Formalin induced nociception assay (120 min, inhibition rate of licking time, %) | | CFA induced nociception assay (pain threshold improvement rate, %) |
|---|---|---|---|---|---|
| | | | First phase | Second phase | 7th compare with 1st |
| Aspirin group (200 mg/kg) | 75 | 19.21 | 50.31 | 25.97 | 152.03 |
| Low dosage AC (0.3 mg/kg) | 68 | 17.12 | 33.23 | 20.25 | 131.33 |
| High dosage AC (0.9 mg/kg) | 76 | 20.27 | 58.92 | 68.32 | 35.11 |

and 0.9 mg/kg of AC were chosen for subsequent studies. At these concentrations, the mice did not show symptoms of poisoning. As shown in Table 3, AC could relieve pain caused by various factors. The hot plate test is often used to establish thermally-induced acute, neuropathic pain model [30, 31]. In this experiment, AC improved the pain threshold of mice by 20.27% at concentration of 0.9 mg/kg, which indicated that AC was able to relieve acute thermal stimulus pain. Besides, the writhing behavior of mice evoked by acetic acid injection is considered to reflect visceral pain [7, 32], and 0.9 mg/kg of AC could reduce the number of writhing by 76%. Therefore, AC has shown significant therapeutic efficacy for visceral pain. In the process of inflammatory pain, AC could improve the pain threshold both in formalin and CFA induced mice pain models. Alternative, in the inflammatory pain caused by formalin, AC treatment showed remarkable inhibition effects on the paw licking time in both primary and secondary phases. Most importantly, the inhibition rate of 0.9 mg/kg AC treatment was 2.6 times higher than aspirin group in the second phase. In the CFA-induced inflammatory pain model, the improvement of pain threshold in AC treatment group was similar to that of aspirin group. Thus, AC treatment also showed powerful inhibitory effect on inflammatory pain. However, the analgesic mechanism of aconitine has not been reported in detail and needs further study.

In summary, the present study demonstrates that the administration of AC, a diester alkaloid, has anti-nociception activities in mice pain models caused by hot plate, acetic acid, formalin and CFA. From this we can infer that AC has therapeutic effect on different types of pain, including acute thermal stimulus pain, visceral pain and inflammatory pain. These results will guide more accurate application of AC clinical analgesia.

## Author Contributions

**Conceptualization:** Elaine Lai-Han Leung.

**Data curation:** Jianhua Deng, Jiada Han, Jiahao Chen, Yanmin Zhang, Qiuju Huang.

**Formal analysis:** Jianhua Deng, Yanmin Zhang.

**Funding acquisition:** Linlin Lu.

**Investigation:** Jiahao Chen, Yanmin Zhang, Xiaoxiao Qi.

**Project administration:** Jianhua Deng, Qiuju Huang, Qian Feng, Linlin Lu.

**Resources:** Xiaoxiao Qi, Zhongqiu Liu, Linlin Lu.

**Software:** Ying Wang, Zhongqiu Liu.

**Supervision:** Zhongqiu Liu, Elaine Lai-Han Leung, Dawei Wang, Linlin Lu.

**Validation:** Jianhua Deng, Jiada Han, Elaine Lai-Han Leung, Qian Feng, Linlin Lu.

**Visualization:** Jianhua Deng, Jiada Han, Qian Feng, Linlin Lu.

**Writing – original draft:** Jianhua Deng, Jiada Han, Qian Feng, Linlin Lu.

**Writing – review & editing:** Qian Feng, Linlin Lu.

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
