## [Decision Letter · Decision Letter 0]

8 Dec 2020

PONE-D-20-32048

Comparison of Analgesic Activities of Aconitine in Different Mice Pain Models

PLOS ONE

Dear Dr. Feng,

Thank you for submitting your manuscript to PLOS ONE. After careful consideration, we feel that it has merit but does not fully meet PLOS ONE’s publication criteria as it currently stands. Therefore, we invite you to submit a revised version of the manuscript that addresses the points raised during the review process.

Please pay careful attention to the issues raised by the reviewers. Both reviewers were highly concerned about the rigor of experimental design and analysis, as well as the organization of the manuscript and the conclusions drawn from the data. A successful revision will take these issues strongly into account.

We look forward to receiving your revised manuscript.

Kind regards,

John M. Streicher, Ph.D.

Academic Editor

PLOS ONE

Journal Requirements:

2. To this end, please revise your Methods section to address the following: (1) the number of animals in each group and how you determined the sample size; (2) the sex and strain of the mice; (3) all anesthetics and analgesics administered to animals during your study (name of drug, dosage, frequency and route of administration); (4) details about humane endpoints for any animals who became severely ill or injured during the study; (5) the rate of mortality during the study and the cause of death (if applicable); (6) the criteria used to determine when to euthanize animals (for animals who became ill prior to the experimental endpoints). Lastly, please complete and submit the ARRIVE Guidelines 2.0 checklist (Essential 10 version): https://arriveguidelines.org/resources/author-checklists .

4. Please include your tables as part of your main manuscript and remove the individual files. Please note that supplementary tables (should remain/ be uploaded) as separate "supporting information" files

Reviewers' comments:

Reviewer's Responses to Questions

**Comments to the Author**

1. Is the manuscript technically sound, and do the data support the conclusions?

Reviewer #1: No

Reviewer #2: Partly

2. Has the statistical analysis been performed appropriately and rigorously? 

Reviewer #1: No

Reviewer #2: I Don't Know

3. Have the authors made all data underlying the findings in their manuscript fully available?

Reviewer #1: No

Reviewer #2: Yes

4. Is the manuscript presented in an intelligible fashion and written in standard English?

Reviewer #1: No

Reviewer #2: No

5. Review Comments to the Author

Reviewer #1: This manuscript by Deng et al. aimed to determine the analgesic activity of aconitine (AC) on thermal and inflammatory pain. The authors performed hot plate test to determine the impact of AC on thermal pain, and then used the acetic acid-, formaldehyde-, and CFA-induced pain to test the effect of AC on inflammatory pain. They found that treatment with AC exhibited analgesic effects in those pain models. However, this is not a well-designed study and a poorly written manuscript.

Major Compulsory Revisions

1) Provide the rationale for this study. What do the results mean?

2) Abstract: provide more specific information about methodology; the conclusion should be re-written.

3) Introduction:

a. The whole section is a lack logic and needs to be organized.

b. The statement addressed in Line 53-55 is incorrect.

c. Add the reference for the statement in Line 61.

4) Methods:

a. The purchased AC needs to be characterized to confirm.

b. Provide more details about how the mice were housed; the temperature and humidity could induce some stress for the mice; the groups of animals are inconsistent (4 or 5 groups), and the name of groups are very confused (vehicle, normal, model) through the whole manuscript.

c. Provide rationale for the dosage of AC since it has been demonstrated serious toxicity; please provide the reference of the authors’ previous study mentioned in Line 232.

d. The description of doses of AC is inconsistent through the methods section.

e. The description of methods is confusing, including Line 115-116, Line 121-123, Line 123-125, and Line 126-127.

f. Please confirm that the formaldehyde and CFA were injected into the toes of mice; how many toes were injected?

g. The impact of AC on the “normal group mice” should be included in the CFA and formaldehyde experiments.

h. Provide the post-comparison method in the statistical analysis.

5) Results:

a. The names of treatment groups are very hard to follow.

b. Please define the “viz” (Line 151) and T1/2 (Line 168).

c. The data from two doses of AC can NOT support any “dose (concentration)-dependent manner” effects, and the impact of the low and high dose of AC was very tight in most of the experiments.

d. The authors mentioned indomethacin in Line 161, where is that from? Please clarify.

e. Descriptions are confused, such as Line 169-170, Line 187-188, Line 188-190.

f. Explain the statement in Line 186.

g. Clarify the method used for determining the mechanical hyperalgesia (Line 194).

h. The whole section from Line 198-218 is unnecessary. If the authors really want to keep it, then it should be merged into the discussion.

i. Improve the quality of images in Figure 2A.

6) Discussion:

a. Why do the readers need to know the information in Line 222-223?

b. The acute treatment with acetic acid is hard to induce any neuropathic pain. It is unclear where the statement is in Line 240-242 from.

c. It is a lack of evidence to support the statement in Line 256-257.

7) The whole manuscript needs to be edited by a native speaker of English.

Reviewer #2: While the findings of this study are interesting, the conclusions that are drawn by the authors are somewhat inappropriate. There are also problems with the language used throughout the manuscript that make it difficult to understand. Specifically, in their conclusions statement of the abstract as well as in their discussion section, they state that ‘the analgesic effect of aconitine (AC) was that pain caused by acetic acid stimulation was greater than pain induced by heat stimuli, and acute inflammatory pain was greater than chronic inflammatory pain’. This statement is very confusing because they begin talking about the analgesic effect of AC and then they describe that pain was greater in some of their models. They should be describing the analgesic effect throughout their conclusion. They should also use the language of the specific assays/models that they used to measure the analgesic effect instead of the broad statement of acute inflammatory pain or chronic inflammatory pain.

Additionally, in their discussion they compare the pain induced by heat stimuli to the pain of acetic acid stimulation and state that AC had stronger analgesic effect on the acetic acid test. This is not an appropriate comparison because the hot plate test is not a model of pain but rather an assay to measure heat withdrawal latency. The acetic acid test is used to evoke pain-like behavior and therefore there is much more room for an analgesic effect. The hot plate test was simply measuring heat withdrawal latency with nothing to evoke pain. The authors also consistently call the formalin model an acute pain model and the CFA model a chronic pain model and this is not entirely accurate. CFA injection does not necessarily cause chronic pain.

As a suggestion for a more appropriate way to draw conclusions, the authors could simply state that AC had an analgesic effect in each of their pain models instead of ranking the analgesic effect and comparing it between the models, as all of the models were measuring different things.

Other comments:

Methods:

In the ‘animals and treatment’ section of the methods the authors state that all animals were divided into 5 groups but it doesn’t state what these groups are.

Were male or female mice combined in the assays? If so, were the sexes compared to see if there were differences between males and females?

The hot plate test is not well described. The authors state ‘the latency of 30 minutes after administration was measured 3 times’ but do not describe what the latency is. Is it latency to withdraw paw? Also, the pain threshold improvement rate equation describes a base threshold but how and when this was measured is not described.

In the formalin test, how were saline, aspirin, and AC given?

In the formalin test, the authors state ‘after 60 and 120 min of administration, formalin was injected’. This language is confusing and makes it seem like the saline, aspirin, and AC were administered continuously for 60 and 120 minutes.

In the formalin test, what is the cumulative time of lameness? Usually in the formalin test the amount of time spent licking the paw is quantified.

There is nothing in the statistics section or in the figure/table legends about post-hoc analyses, which is where it appears the significance comes from.

Results:

The authors state in the formalin-induced paw licking model section: ‘AC administration for 120 minutes also significantly inhibited formalin-induced paw withdrawal time in a concentration-dependent manner’. Should this say licking time? Or was something else measured?

Edema is not a measure of pain but a measure of swelling, which is often associated with inflammation and commonly measure following CFA injection. This language should be changed to reflect this.

The authors state that ‘this-CFA induced mechanical hyperalgesia was inhibited by administration with AC’, however they are measuring thermal hyperalgesia.

Why in the CFA model are the mice treated multiple times with AC when in the other tests they are only treated once?

Discussion

The sentence ‘As far as we know, AC has a wide range of analgesic properties, but whether there is a difference in the treatment among pains by AC, and which kind of pain has the best

therapeutic effect have not been demonstrated in reports.’ Is very confusing and not worded well.

The statement about the concentration of AC that was previously found should be in the methods section somewhere.

There could be some discussion as to why in the CFA model, the lesser dose of AC seemed to work better than the higher dose of AC but the higher dose of AC worked better in the formalin model and the writhing pain and hot plate (slightly)

Figures and tables:

The figures and tables 1 & 2 look nice and are organized well. The tables would be more compelling as figures.

The figure 1 legend says the administration of AC or aspirin for 60 (or 120) minutes- does this mean given 60 minutes before?

In Figure 2, were there actual measurements of edema taken or just pictures?

In the Figure 2A legend, they mention the analgesic effect of aconitine but only edema is shown, not analgesia

In Figure 2 need to state what the test is in Figure 2B- says contraction latency but not from the hotplate test

6. PLOS authors have the option to publish the peer review history of their article (what does this mean?). If published, this will include your full peer review and any attached files.

Reviewer #1: No

Reviewer #2: No

---

## [Author Response · Author response to Decision Letter 0]

22 Jan 2021

Jan 21th, 2021

John M. Streicher, Ph.D.

Academic Editor

PLOS ONE

Dear Dr. Streicher,

Thank you very much for your consideration of our manuscript entitled “Comparison of Analgesic Activities of Aconitine in Different Mice Pain Models” (ID: PONE-D-20-32048) to be published in PLOS ONE. Also, we gratefully appreciate for all of the suggestions and comments from reviewers. Those comments are all valuable and very helpful for revising and improving our paper. We have studied comments carefully and have made corrections which we hope meet with approval. We have modified the article in revisions mode. Meanwhile, all the figures and tables were restyled according to the corresponding guidelines. Our point-to-point responses to editor and the reviewers’ comments are shown below. We hope that the revised paper is sufficient for publication in PLOS ONE.

Best wishes!

Regards,

Qian Feng, Ph.D. 

International Institute for Translational Chinese Medicine, 

Guangzhou University of Chinese Medicine, 

Guangzhou, 510006, China

Tel: +86-020-39358071; Email: fengqian@gzucm.edu.cn

 

Comments from the editors and reviewers:

-Reviewer 1

This manuscript by Deng et al. aimed to determine the analgesic activity of aconitine (AC) on thermal and inflammatory pain. The authors performed hot plate test to determine the impact of AC on thermal pain, and then used the acetic acid-, formaldehyde-, and CFA-induced pain to test the effect of AC on inflammatory pain. They found that treatment with AC exhibited analgesic effects in those pain models. However, this is not a well-designed study and a poorly written manuscript. 

Major Compulsory Revisions

1) Provide the rationale for this study. What do the results mean?

Response: Thank you so much for your constructive suggestion. It is well known that aconitine (AC) is a secondary metabolite alkaloid and main bioactive compound in Aconitum plants, which accounting for more than 60% of the total diester-diterpenoid alkaloids. AC is a complex compound, which not only has a variety of biological activities, but also shows certain toxic and side effects. The previous study of our research also confirmed that AC has anti-inflammatory, analgesic, anti-tumor and other biological activities. The main aim of this study is to investigate the analgesic effect of AC in vivo and compare the analgesic efficacy in different mice pain models. Herein, we established four different mice pain models including hot plate test, acetic acid-, formaldehyde-, and CFA-induced pain models. Our results showed that in hot plate experiment, AC at concentration of 0.9 mg/kg significantly prolonged the paw withdrawal latency (3.0±1.35 VS 7.6±1.42), which is higher than 200 mg/kg of Aspirin (5.0±1.42). In the acetic acid writhing test, AC significantly reduced writhing times from 28.83±11.53 to 7±5.14 in 60 minutes. In formaldehyde induced pain model, AC not only significantly reduced formalin-induced paw licking time in the first phase (107±43.56 vs 62±25.04), but also showed remarkable analgesic effect in second phase. Remarkably, the analgesic effect of 0.9 mg/kg AC was 2.6 times better than 200 mg/kg aspirin in the second phase. At last, we also found that after 7 days AC treatment at the concentration of 0.3 mg/kg, mice pain threshold improvement rate was 131.33%. Our results remind that AC treatment not only could inhibit thermally-induced neuropathic pain, but also had well analgesic effect on inflammatory pain. This study will provide some evidences for the accurate application of AC in clinical treatment.

2) Abstract: provide more specific information about methodology; the conclusion should be re-written.

Response: Thank you so much for your great efforts in improving the quality of our manuscript. The whole “Abstract” section has been re-written in our revised manuscript (Line 23-39).

3) Introduction:

a. The whole section is a lack logic and needs to be organized.

Response: Thank you so much for your patience and carefulness on our manuscript. The whole “Introduction” section has been reorganized in our revised manuscript (Line 41-92). 

b. The statement addressed in Line 53-55 is incorrect.

Response: The incorrect statement addressed in Line 53-55 has been corrected in our revised manuscript (Line 53-54). 

c. Add the reference for the statement in Line 61.

Response: The reference of Line 61 has been added in the revised manuscript (Walters ET, et al. 2019).

4) Methods:

a. The purchased AC needs to be characterized to confirm.

Response: Very appreciated for your constructive suggestion. Actually, the compound of AC was purchased from Chengdu Must Bio-Technology Co., Ltd. (Chengdu, China). The purity of AC is greater than 98% tested by HPLC assay. The analysis certificate provided by manufacturer was added as below. 

b. Provide more details about how the mice were housed; the temperature and humidity could induce some stress for the mice; the groups of animals are inconsistent (4 or 5 groups), and the name of groups are very confused (vehicle, normal, model) through the whole manuscript.

Response: Very appreciated for your important advice. The detail environments of mice were added in the “Material and Methods” section (Line 108-110) in our revised manuscript. The names of groups were normalized as control (0.9% saline treatment), model (formaldehyde- or CFA-induced), aspirin treatment (200 mg/kg administration), AC treatment (0.3 mg/kg or 0.9 mg/kg administration).

c. Provide rationale for the dosage of AC since it has been demonstrated serious toxicity; please provide the reference of the authors’ previous study mentioned in Line 232.

Response: Thank you so much for your insightful suggestion. Indeed, AC not only has a variety of biological activities, but also shows certain toxic effects. Our research group has done plenty work of AC including the absorption and metabolism processes of AC in mice, and we found that the LD50 of AC in mice was 1.8 mg/kg (Lijun Zhu, et al. 2017; Fan He, et al. 2017; Xiaocui Li, et al. 2020). Meanwhile, Wada K (Wada K, et al. 2006) also found that mice were administrated orally with AC at the concentration of 1.0 mg/kg for 21 days did not show obviously toxic effects. Therefore, the dosages of AC in this study were chosen as 0.3 mg/kg and 0.9 mg/kg. Most importantly, the longest administration of AC in our study was no more than 7 days. During the whole experiment, there were no obviously toxic effects like vomiting and convulsion were observed.

d. The description of doses of AC is inconsistent through the methods section.

Response: Thank you for your patience and carefulness on our manuscript. We have corrected this mistake in the revised manuscript (Line 117-118, Line 127-128, Line137-138 and Line147-148). In this study, two different dosages of AC including 0.3 mg/kg (low dose group) and 0.9 mg/kg (high dose group) were used in the experiment. 

e. The description of methods is confusing, including Line 115-116, Line 121-123, Line 123-125, and Line 126-127.

Response: Thank you so much for your detail but constructive suggestions, we have corrected these confusing descriptions in our revised manuscript (Line 119-120, Line 128-130, Line 131-133). 

f. Please confirm that the formaldehyde and CFA were injected into the toes of mice; how many toes were injected?

Response: Thank you very much for your suggestion. Maybe there were some mistakes in the description, actually, formaldehyde and CFA were subcutaneously injected into the dorsal surface of the right hindpaw of mice. We have corrected the mistakes in our revised manuscript (Line 139-141, 148-150).

g. The impact of AC on the “normal group mice” should be included in the CFA and formaldehyde experiments.

Response: Very appreciated for your knowledgeable suggestion. Indeed, it is better to explore the impact of AC on the control mice to compare the effects of AC in normal mice and CFA/formaldehyde induced mice. In our study, we want to investigate the analgesic effect of AC in different mice models, so we just tested the effect of AC in CFA/formaldehyde induced mice. In the future studies, we will pay more attention to this phenomenon.

h. Provide the post-comparison method in the statistical analysis.

Response: Thank you so much for your suggestion. The detail information of statistical analysis was added in our revised manuscript. In particular, data were expressed as the mean ± standard deviation (SD). Significant differences were analyzed using one-way analysis of variance (ANOVA) followed by Tukey’s post hoc test for multiple comparisons by SPSS 20.0. Statistical significances were considered at p < 0.05 (Line 158-161).

5) Results:

a. The names of treatment groups are very hard to follow.

Response: Thanks again for your patience. The names of groups were normalized in our revised manuscript in the “Material and Methods” section (Line 94-160).

b. Please define the “viz” (Line 151) and T1/2 (Line 168).

Response: Very appreciated for your carefulness on our manuscript. ”viz.” is short for “videlicet” and T1/2 is short for half-life. T1⁄2 is the time required for a quantity to reduce to half of its initial value. The medical sciences refer to the biological half-life of drugs and other chemicals in the human body. The original term, half-life period, dating to Ernest Rutherford's discovery of the principle in 1907, was shortened to half-life in the early 1950s (John Ayto, 1989).

c. The data from two doses of AC can NOT support any “dose (concentration)-dependent manner” effects, and the impact of the low and high dose of AC was very tight in most of the experiments.

Response: Thank you so much for your constructive suggestion. Indeed, it is impossible to support “dose (concentration)-dependent manner” with only two concentrations. Our original intention is to explain both high and low doses of AC have analgesic effects. This inaccurate description has been corrected on the revised manuscript. Thanks again for your patience.

d. The authors mentioned indomethacin in Line 161, where is that from? Please clarify.

Response: Very appreciated for your carefulness on our manuscript. This is our mistake. Actually, the positive compound in acetic acid-induced writhing test was “aspirin” instead of “indomethacin”, we have corrected the description in the revised manuscript.

e. Descriptions are confused, such as Line 169-170, Line 187-188, Line 188-190.

Response: Thank you so much for your great efforts in improving the quality of our manuscript. we have corrected these descriptions in Line 200-202, Line 227-229 and Line 229-230 in the revised manuscript. Thanks again for your patience and carefulness.

f. Explain the statement in Line 186.

Response: The statement in Line 186 is “saline was injected into the normal group, the swelling of the plantar of mice was observed only on day 1” which means the swelling of mice paw induced by solvent is temporary and do not affect the analgesic observation of AC subsequently. This is a sham operation group and used to compare with CFA-induced modeling group.

g. Clarify the method used for determining the mechanical hyperalgesia (Line 194).

Response: In our study, we used the hot plate test to assess thermal hyperalgesia as mentioned in the previous article (Calixto-Campos C, 2015;), but mechanical hyperalgesia was not measured. The description in Line 194 about “mechanical hyperalgesia” is not correct, we have revised it to “thermal hyperalgesia” (Line 232-233).

 h. The whole section from Line 198-218 is unnecessary. If the authors really want to keep it, then it should be merged into the discussion.

Response: Thank you so much for your insightful suggestion. The whole section of Line 198-218 has been modified and removed to the discussion section (Line 247-286). 

i. Improve the quality of images in Figure 2A.

Response: Thank you so much and we resubmitted the Figure 2 with the resolution more than 300 ppi.

6) Discussion:

a. Why do the readers need to know the information in Line 222-223?

Response: Thank you very much for your important question. The information tells us AC has great clinical value which is widely application and definite efficacy. Therefore, it is meaningful to explore the analgesic effect of AC.

b. The acute treatment with acetic acid is hard to induce any neuropathic pain. It is unclear where the statement is in Line 240-242 from.

Response: Thank you so much for your constructive suggestion. Indeed, the writhing behavior of mice evoked by acetic acid is well known to reflect visceral pain (Gawade SP, 2012; Wada K, 2006). The incorrect description in the manuscript has been modified in the revised manuscript. (Line 271)

c. It is a lack of evidence to support the statement in Line 256-257.

Response: Very appreciated for your carefulness on our manuscript. As your mentioned, it is inappropriate to compare the analgesic effect of AC between different pain models. In the revised manuscript, we just analyzed the analgesic effect of AC in four different pain models.

7) The whole manuscript needs to be edited by a native speaker of English.

Response: Many thanks for your great efforts in improving the quality of our manuscript. The manuscript has been edited for proper English language, grammar, punctuation, spelling, and overall style by one or more of the highly qualified native English speakers. 

 

-Reviewer 2

While the findings of this study are interesting, the conclusions that are drawn by the authors are somewhat inappropriate. There are also problems with the language used throughout the manuscript that make it difficult to understand. Specifically, in their conclusions statement of the abstract as well as in their discussion section, they state that ‘the analgesic effect of aconitine (AC) was that pain caused by acetic acid stimulation was greater than pain induced by heat stimuli, and acute inflammatory pain was greater than chronic inflammatory pain’. This statement is very confusing because they begin talking about the analgesic effect of AC and then they describe that pain was greater in some of their models. They should be describing the analgesic effect throughout their conclusion. They should also use the language of the specific assays/models that they used to measure the analgesic effect instead of the broad statement of acute inflammatory pain or chronic inflammatory pain.

Additionally, in their discussion they compare the pain induced by heat stimuli to the pain of acetic acid stimulation and state that AC had stronger analgesic effect on the acetic acid test. This is not an appropriate comparison because the hot plate test is not a model of pain but rather an assay to measure heat withdrawal latency. The acetic acid test is used to evoke pain-like behavior and therefore there is much more room for an analgesic effect. The hot plate test was simply measuring heat withdrawal latency with nothing to evoke pain. The authors also consistently call the formalin model an acute pain model and the CFA model a chronic pain model and this is not entirely accurate. CFA injection does not necessarily cause chronic pain.

As a suggestion for a more appropriate way to draw conclusions, the authors could simply state that AC had an analgesic effect in each of their pain models instead of ranking the analgesic effect and comparing it between the models, as all of the models were measuring different things.

Response: Very appreciate for your constructive suggestion, which is very important to us. We fully agree that it’s inappropriate to rank the analgesic effect of AC on different pain models due to the models induced by different methods represent different kinds of pain. In the revised manuscript, we only demonstrated that AC had analgesic effect in four different pain models including hot plate test, acetic acid-, formaldehyde-, and CFA-induced pain models.

Other comments:

1) Methods:

a. In the ‘animals and treatment’ section of the methods the authors state that all animals were divided into 5 groups but it doesn’t state what these groups are.

Response: Thank you very much for your important question. The five groups included control (0.9% saline treatment), model (formaldehyde- or CFA-induced), aspirin treatment (200 mg/kg administration), AC treatment (0.3 mg/kg or 0.9 mg/kg administration). The details were added in the “Material and Methods” section (Line 135-154) in our revised manuscript.

b. Were male or female mice combined in the assays? If so, were the sexes compared to see if there were differences between males and females?

Response: Thank you very much for your thought-provoking question. All the mice used in the experiment were female. We only compared the efficacy of a single gender, and we will consider adding different genders for comparison later.

c. The hot plate test is not well described. The authors state ‘the latency of 30 minutes after administration was measured 3 times’ but do not describe what the latency is. Is it latency to withdraw paw? Also, the pain threshold improvement rate equation describes a base threshold but how and when this was measured is not described.

Response: Thank you very much for your constructive question, we have already added detail descriptions in the “Material and Methods” section in our revised manuscript (Line 115-123). Response time for hindpaw withdraw of mice was recorded 30 min after the administration of compounds. The base threshold of hindpaw withdraw latency was detected 30 min before the administration of compounds.

d. In the formalin test, how were saline, aspirin, and AC given?

Response: Very appreciated for your carefulness on our manuscript. Actually, in the whole experiment, saline, aspirin and different concentrations of AC were administrated orally.

e. In the formalin test, the authors state ‘after 60 and 120 min of administration, formalin was injected’. This language is confusing and makes it seem like the saline, aspirin, and AC were administered continuously for 60 and 120 minutes.

Response: Many thanks for your constructive question. Maybe, there are some unclear descriptions in the manuscript which confusing you. Actually, we meant that formalin was used to create inflammatory pain model. After oral administration of AC at 60 and 120 min, mice were modeling with formalin respectively. The analgesic effect of AC on formalin induced acute tissue injury pain was observed within 1 hour after formalin injection. The first 5 min observation is defined as the first phase, and the 15 to 60 min is called the second phase. 

f. In the formalin test, what is the cumulative time of lameness? Usually in the formalin test the amount of time spent licking the paw is quantified.

Response: Thank you so much for your insightful question. As a matter of fact, we measured the cumulative time of paw licking in the formalin test. We have corrected the mistake.

g. There is nothing in the statistics section or in the figure/table legends about post-hoc analyses, which is where it appears the significance comes from.

Response: Thank you very much for your suggestion. According to your suggestion, we have provided the detail information about post-hoc analyses in the “Material and Methods-Statistical analysis” section in our revised manuscript (Line 156-160). Thanks again for improving the quality of our manuscript. 

2) Results:

a. The authors state in the formalin-induced paw licking model section: ‘AC administration for 120 minutes also significantly inhibited formalin-induced paw withdrawal time in a concentration-dependent manner’. Should this say licking time? Or was something else measured?

Response: Thank you very much for your important question. In formaldehyde induced pain model, AC significantly reduced formalin-induced paw licking time in both first and second phases. The incorrect description about “paw withdrawal time” in the manuscript has been modified in the revised manuscript (Line 208-210). 

b. Edema is not a measure of pain but a measure of swelling, which is often associated with inflammation and commonly measure following CFA injection. This language should be changed to reflect this.

Response: Thank you so much for your constructive suggestion. Under treatment with 0.3 mg/kg or 0.9 mg/kg of AC, paw edema of mice was effectively suppressed, indicating that AC relieved the inflammation caused by CFA. We have revised the section in Line 229-231. 

c. The authors state that ‘this-CFA induced mechanical hyperalgesia was inhibited by administration with AC’, however they are measuring thermal hyperalgesia.

Response: Thank you so much for your carefulness on our manuscript. In our study, we used the hot plate test to assess thermal hyperalgesia. The description of “mechanical hyperalgesia” is not appropriate, we have revised it to “thermal hyperalgesia”. 

d. Why in the CFA model are the mice treated multiple times with AC when in the other tests they are only treated once?

Response: Thank you very much for your constructive question. According to previous studies, CFA-induced pain model was often used to study persistent and chronic inflammatory pain. The inflammatory pain caused by CFA could last up to 16 days [Wandji BA, 2018; Britch SC, 2020; Piegang BN, 2020]. In this study, we conducted CFA model to investigate the analgesic effect of AC on persistent inflammatory pain for 7 days. However, as our previous study, we know that AC is metabolized quickly in mice, and the T1/2 of AC in mice is about 2 hours [Zhang M, 2015], indicating that a single administration of AC failed to achieve long-term analgesic effect. Above all, in CFA induced pain model, mice were administrated with AC once a day for 7 days to ensure the long-term analgesic effect.

3) Discussion

a. The sentence ‘As far as we know, AC has a wide range of analgesic properties, but whether there is a difference in the treatment among pains by AC, and which kind of pain has the best therapeutic effect have not been demonstrated in reports.’ Is very confusing and not worded well.

Response: Thanks very much for improving the quality of our manuscript. As we all known that AC has a wide range of biological activities like anti-inflammatory, analgesic, anti-tumor and so on according to our previously studies or other researchers. This study we investigated the analgesic effect of AC in vivo and the original aim was to compare the different among different pain models. Latterly, both our results and comments of reviewers told us that different pain models could not be compared due to the different pathogenesis. Therefore, we apologize for the confusion caused by our unclear descriptions. We have already corrected these mistakes in our revised manuscript. Thanks again for your patience.

b. The statement about the concentration of AC that was previously found should be in the methods section somewhere.

Response: Very appreciated your constructive suggestion and the reference was added in our revised “Discussion” section (Line 261-262)

c. There could be some discussion as to why in the CFA model, the lesser dose of AC seemed to work better than the higher dose of AC but the higher dose of AC worked better in the formalin model and the writhing pain and hot plate (slightly)

Response: Many thanks for your insightful question. According to Figure 2B, the paw withdrawal latency of mice at the high dosage AC (0.9 mg/kg) group was longer than that of the low dosage AC (0.3 mg/kg) group on the 1st and 3rd day. However, after the 5th day, the paw withdrawal latency of the high dosage AC (0.9 mg/kg) group was shorter than that of the low dosage AC (0.3 mg/kg) group, but there was no significant difference (p>0.05). The reason why the treatment effect of low dosage AC was better than that of high dosage AC after long time treatment was unclear. At present, we do not know whether the repeated administration of 0.9 mg/kg of AC will cause the production of pain-related factors, thereby affecting the analgesic effect of AC. We will further study the analgesic effect of AC.

4) Figures and tables:

a. The figures and tables 1 & 2 look nice and are organized well. The tables would be more compelling as figures.

b. The figure 1 legend says the administration of AC or aspirin for 60 (or 120) minutes- does this mean given 60 minutes before?

c. In Figure 2, were there actual measurements of edema taken or just pictures?

d. In the Figure 2A legend, they mention the analgesic effect of aconitine but only edema is shown, not analgesia

e. In Figure 2 need to state what the test is in Figure 2B says contraction latency but not from the hotplate test

Response: Many thanks for your great efforts in improving the quality of our manuscript. Indeed, figures are more compelling than tables, in the future, we will pay more attention to this phenomenon, in order to present our results more compelling. As well, we revised the figure legends of figure 1 and 2. In Figure 2A, we only photographed the paw swelling of mice, we are more focused on the analgesic effect of AC which reflected in the measurement of thermal hyperalgesia in mice. Thanks again for your carful checking on our manuscript. 

-Managing Editor

Response: Thank you very much for your suggestion, we have restyled our manuscript including manes of files according to the guidelines.

2. To this end, please revise your Methods section to address the following: (1) the number of animals in each group and how you determined the sample size; (2) the sex and strain of the mice; (3) all anesthetics and analgesics administered to animals during your study (name of drug, dosage, frequency and route of administration); (4) details about humane endpoints for any animals who became severely ill or injured during the study; (5) the rate of mortality during the study and the cause of death (if applicable); (6) the criteria used to determine when to euthanize animals (for animals who became ill prior to the experimental endpoints). Lastly, please complete and submit the ARRIVE Guidelines 2.0 checklist (Essential 10 version): https://arriveguidelines.org/resources/author-checklists .

Response: Thank you so much for your suggestion, we have revised our “Material and Methods” section. Besides, the ARRIVE Guidelines 2.0 checklist (Essential 10 version) has been submitted with revised manuscript.

Response: Very appreciated for your suggestion, we have already had an ORCID ID for the corresponding author.

4. Please include your tables as part of your main manuscript and remove the individual files. Please note that supplementary tables (should remain/ be uploaded) as separate "supporting information" files

Response: Many thanks for your suggestion. We have already added our tables as part of our revised manuscript and removed the individual files.

Response: Thank you so much for your suggestion. Our ethics statement is written in “Material and Methods” section as requested.

---

## [Decision Letter · Decision Letter 1]

8 Feb 2021

PONE-D-20-32048R1

Comparison of Analgesic Activities of Aconitine in Different Mice Pain Models

PLOS ONE

Dear Dr. Feng,

Thank you for submitting your manuscript to PLOS ONE. After careful consideration, we feel that it has merit but does not fully meet PLOS ONE’s publication criteria as it currently stands. Therefore, we invite you to submit a revised version of the manuscript that addresses the points raised during the review process.

We look forward to receiving your revised manuscript.

Kind regards,

John M. Streicher, Ph.D.

Academic Editor

PLOS ONE

Additional Editor Comments (if provided):

Thank you for your revised submission. Both Reviewers still had minor comments that need to be addressed. If you can address these, I can make the final decision without further rounds of review. Thank you!

Reviewers' comments:

Reviewer's Responses to Questions

**Comments to the Author**

1. If the authors have adequately addressed your comments raised in a previous round of review and you feel that this manuscript is now acceptable for publication, you may indicate that here to bypass the “Comments to the Author” section, enter your conflict of interest statement in the “Confidential to Editor” section, and submit your "Accept" recommendation.

Reviewer #1: (No Response)

Reviewer #2: (No Response)

2. Is the manuscript technically sound, and do the data support the conclusions?

Reviewer #1: (No Response)

Reviewer #2: Partly

3. Has the statistical analysis been performed appropriately and rigorously? 

Reviewer #1: Yes

Reviewer #2: Yes

4. Have the authors made all data underlying the findings in their manuscript fully available?

Reviewer #1: Yes

Reviewer #2: Yes

5. Is the manuscript presented in an intelligible fashion and written in standard English?

Reviewer #1: No

Reviewer #2: Yes

6. Review Comments to the Author

Reviewer #1: The manuscript has been improved in the revised version. However, several major issues remain to be improved.

Major Compulsory Revisions

1) The authors only changed the format of the abstract but not really address the comments brought up in the primary evaluation.

2) Introduction: This section is still not logically described and needed to be improved.

3) Methods: The “model” group is the same treatment as the “control” group based on the information in Line 137-138 and Line 147-148.

4) Results: The low dose of AC showed the most significant improvement based on the images of the hind paw in Figure 2. However, the bar graph doesn’t indicate that.

5) The English language issue still needs to be improved throughout the whole manuscript.

Reviewer #2: The authors have appropriately addressed my original comments but the revised version has a few things that need to be addressed.

1. In the abstract (line 27) the authors state they are looking at the analgesic activities of AC on neuropathic pain. However, they do not use a neuropathic pain model. This should be revised to say they are looking at the analgesic effects of AC on thermal sensitivity, or something similar.

2. Similarly, in the abstract (line 38) they state AC had analgesic effect on neuropathic pain but they only use the hot plate to measure thermal sensitivity in naive mice, not a neuropathic pain model. Please revise this sentence as well.

3. The use of the word 'videlicet' in line 169 does not seem appropriate

4. The sentence ending on Line 236 would make more sense if it read '48 hours after CFA injection' instead of 'after CFA injection 48 hours'

5. The conclusion drawn in lines 236-238 is not accurate unless the the different time points following CFA injection were compared statistically. From figure 2, it does not appear that the different time points were compared, only the different groups.

6. The wording of the conclusion on line 267 that AC was able to relieve neuropathic pain is inappropriate (similar to comments 1 and 2). This should say thermal sensitivity or something similar.

7. Lines 282-283 also mention AC having an effect on neuropathic pain

8. In Table 3 for the CFA induced nociception assay, what does the 7th compare with 1st value mean?

7. PLOS authors have the option to publish the peer review history of their article (what does this mean?). If published, this will include your full peer review and any attached files.

Reviewer #1: No

Reviewer #2: No

---

## [Author Response · Author response to Decision Letter 1]

13 Mar 2021

Mar 13th, 2021

John M. Streicher, Ph.D.

Academic Editor

PLOS ONE

Dear Dr. Streicher,

Thank you very much for your consideration of our manuscript entitled “Comparison of Analgesic Activities of Aconitine in Different Mice Pain Models” (ID: PONE-D-20-32048) to be published in PLOS ONE. Also, we gratefully appreciate for all of the suggestions and comments from reviewers. Those comments are all valuable and very helpful for revising and improving our paper. We have studied comments carefully and have made corrections which we hope meet with approval. We have modified the article in revisions mode. Our point-to-point responses to editor and the reviewers’ comments are shown below. We hope that the revised paper is sufficient for publication in PLOS ONE.

Best wishes!

Regards,

Qian Feng, Ph.D. 

International Institute for Translational Chinese Medicine, 

Guangzhou University of Chinese Medicine, 

Guangzhou, 510006, China

Tel: +86-020-39358071; Email: fengqian@gzucm.edu.cn

 

Comments from the reviewers:

-Reviewer 1

The manuscript has been improved in the revised version. However, several major issues remain to be improved.

Major Compulsory Revisions

(1) The authors only changed the format of the abstract but not really address the comments brought up in the primary evaluation.

Response: Thank you so much for your great efforts in improving the quality of our manuscript. The “Abstract” section has been re-written in our revised manuscript (Line 23-43).

(2) Introduction: This section is still not logically described and needed to be improved.

Response: Very appreciated for your important advice. The “Introduction” section has been reorganized in our revised manuscript (Line 45-99). 

(3) Methods: The “model” group is the same treatment as the “control” group based on the information in Line 137-138 and Line 147-148.

Response: Thank you for your patience and carefulness on our manuscript. (1) In formalin induced nociception assay, mice were randomly divided into high dosage AC, low dosage AC, aspirin, model and control groups, with 6 mice in each group. The mice of high dosage AC, low dosage AC and aspirin groups were administered with 0.9 mg/kg AC, 0.3 mg/kg AC and 200 mg/kg aspirin respectively, while the mice of model and control groups were treated with 0.9% saline. One or two hours after administration, 20 μL of 0.92% formaldehyde solution was subcutaneously injected into the dorsal surface of the right hindpaw of the mice of high dosage AC, low dosage AC, aspirin and model groups, expect that the control group was treated with 0.9% saline.

(2) In CFA induced nociception assay, mice were randomly divided into high dosage AC, low dosage AC, aspirin, model and control groups, with 6 mice in each group. 20 μL of CFA was subcutaneously injected into the dorsal surface of the right hindpaw of the mice of high dosage AC, low dosage AC, aspirin and model groups, while the control group was treated with 0.9% saline instead. After 48 hours, the mice of high dosage AC, low dosage AC and aspirin groups were administered with 0.9 mg/kg AC, 0.3 mg/kg AC and 200 mg/kg aspirin respectively，and the mice of model and control groups were treated with 0.9% saline.

(4) Results: The low dose of AC showed the most significant improvement based on the images of the hind paw in Figure 2. However, the bar graph doesn’t indicate that

Response: Thank you very much for your constructive question. The swelling improvement effect of the 0.3mg/kg AC group was similar to that of the aspirin group, but better than the 0.9mg/kg AC group, which may be due to individual differences. At present, we do not know why low dosage AC was better than high dosage AC after long time treatment. We will further study the analgesic and anti-swelling effects of AC.

(5) The English language issue still needs to be improved throughout the whole manuscript. 

Response: Many thanks for your great efforts in improving the quality of our manuscript. The manuscript has been further edited by highly qualified native English speakers.

-Reviewer 2

The authors have appropriately addressed my original comments but the revised version has a few things that need to be addressed.

(1) In the abstract (line 27) the authors state they are looking at the analgesic activities of AC on neuropathic pain. However, they do not use a neuropathic pain model. This should be revised to say they are looking at the analgesic effects of AC on thermal sensitivity, or something similar.

Response: Thank you so much for your constructive suggestion. The description about “neuropathic pain” has been revised to “acute thermal stimulus pain” (Line 34).

2. Similarly, in the abstract (line 38) they state AC had analgesic effect on neuropathic pain but they only use the hot plate to measure thermal sensitivity in naive mice, not a neuropathic pain model. Please revise this sentence as well.

Response: Thank you so much for your carefulness on our manuscript. This sentence has been revised in Line 34.

3. The use of the word 'videlicet' in line 169 does not seem appropriate

Response: Very appreciated for your great efforts in improving the quality of our manuscript. We have rewritten the sentence in Lin 176-178.

4. The sentence ending on Line 236 would make more sense if it read '48 hours after CFA injection' instead of 'after CFA injection 48 hours'

Response: Many thanks for your helpful suggestion. We have modified the expression in Line 239.

5. The conclusion drawn in lines 236-238 is not accurate unless the different time points following CFA injection were compared statistically. From figure 2, it does not appear that the different time points were compared, only the different groups.

Response: Thank you very much for your important question. Indeed, we didn’t compare the different time points in Figure 2. The conclusion in Line 236-238 has been revised.

6. The wording of the conclusion on line 267 that AC was able to relieve neuropathic pain is inappropriate (similar to comments 1 and 2). This should say thermal sensitivity or something similar.

Response: Thanks very much for improving the quality of our manuscript. The description has been revised to “acute thermal stimulus pain” (Line 269).

7. Lines 282-283 also mention AC having an effect on neuropathic pain

Response: Thank you again for your important suggestion. The description has been revised in Line 284-286.

8. In Table 3 for the CFA induced nociception assay, what does the 7th compare with 1st value mean? 

Response: Thank you for your constructive question. In CFA induced nociception assay, drug-treated groups were orally given AC or aspirin respectively once a day for seven consecutive days. Compared with day 1, the improvement rate of pain threshold on day 7 reflected the analgesic effect of aspirin and AC after the whole treatment cycle. Importantly, the improvement of pain threshold in low dosage AC group was similar to that of aspirin group, which indicating that AC treatment showed powerful inhibitory effect on inflammatory pain.

---

## [Editor Report · Decision Letter 2]

16 Mar 2021

Comparison of Analgesic Activities of Aconitine in Different Mice Pain Models

PONE-D-20-32048R2

Dear Dr. Feng,

We’re pleased to inform you that your manuscript has been judged scientifically suitable for publication and will be formally accepted for publication once it meets all outstanding technical requirements.

Kind regards,

John M. Streicher, Ph.D.

Academic Editor

PLOS ONE
---

## [Editor Report · Acceptance letter]

23 Mar 2021

PONE-D-20-32048R2 

Comparison of analgesic activities of aconitine in different mice pain models 

Dear Dr. Feng:

I'm pleased to inform you that your manuscript has been deemed suitable for publication in PLOS ONE. Congratulations! Your manuscript is now with our production department. 

Kind regards, 

on behalf of

Dr. John M. Streicher 

Academic Editor

PLOS ONE